# Malnutrition and Fall Risk in Older Adults: A Comprehensive Assessment Across Different Living Situations

**DOI:** 10.3390/nu16213694

**Published:** 2024-10-30

**Authors:** Marzanna Mziray, Karolina Nowosad, Aleksandra Śliwińska, Mateusz Chwesiuk, Sylwia Małgorzewicz

**Affiliations:** 1Department of Public Nursing and Health Promotion, Medical University of Gdańsk, 80-210 Gdańsk, Poland; marzanna.mziray@gumed.edu.pl; 2Department of Biotechnology, Microbiology and Human Nutrition, University of Life Sciences in Lublin, 20-400 Lublin, Poland; karolina.nowosad@up.lublin.pl; 3Department of Clinical Nutrition, Medical University of Gdańsk, 80-210 Gdańsk, Poland; aleksandra.sliwinska@gumed.edu.pl (A.Ś.); mchwesiuk@gumed.edu.pl (M.C.)

**Keywords:** malnutrition, assessment of nutritional status, fall risk assessment, older people

## Abstract

Background: Malnutrition among older adults is associated with numerous adverse effects, including increased morbidity, mortality, prolonged hospital stays, and a heightened risk of falls. This study aims to investigate the prevalence of malnutrition in different groups of older adults using the F-MNA, anthropometry, and s-albumin and the association between nutritional status and fall risk. Methods: A total of 228 participants aged 60 years and older were divided into three groups: (1) patients in an internal medicine ward, (2) individuals living in family homes, and (3) residents of care homes. Disease profiles, nutritional status (assessed using the F-MNA and SNAQ), body composition, fall risk, and biochemical markers were evaluated. Results: The results indicated the highest prevalence of malnutrition among hospitalized individuals. Fall risk was associated with age, calf circumference, the F-MNA, the SNAQ, serum albumin levels, residence in a care home, comorbidities, and the number of medications taken daily. Regression analysis revealed that age, calf circumference, and residence in a care home were independent predictors of fall risk in older adults. Conclusion: Older adults are at significant risk of malnutrition, with the risk notably increasing during hospitalization and long-term stays in care homes. Hospitalized individuals had the poorest nutritional status and were at significant risk of further weight loss, underscoring the importance of post-discharge care and rehabilitation.

## 1. Introduction

Malnutrition is a common condition, particularly among older adults, with its incidence in this population reaching up to 60% [1]. Despite this high prevalence, malnutrition is often underestimated in diagnostic and therapeutic procedures due to a lack of awareness among healthcare professionals and caregivers about patients’ nutritional issues [2,3]. Numerous studies have demonstrated a strong correlation between malnutrition and an increased risk of morbidity and mortality in hospitalized older adults [4,5]. Malnutrition is also associated with longer hospital stays, being up to 90% longer compared to those for well-nourished individuals, which consequently leads to higher treatment costs [6,7]. Malnutrition is closely linked to impaired muscle and respiratory function, a weakened immune response, and a general increase in disease incidence and complications [8]. In particular, inflammation elevates the risk of malnutrition in older adults by reducing appetite and increasing catabolism [9].

Malnutrition often coexists with geriatric syndromes, such as frailty, and contributes to an increased risk of falls. Falls pose a significant issue for older adults, with one-third of individuals over the age of 65 experiencing at least one fall per year [10]. Falls increase the risk of mortality and morbidity, negatively impact daily functionality, and are associated with a decline in quality of life. Furthermore, the fear of falling can create a negative cycle in which reduced physical activity increases the need for external assistance with everyday tasks [11,12]. Several chronic conditions have been identified as being associated with a higher incidence of falls in older adults, including Parkinson’s disease [13], heart failure [14], diabetes [15], depression [16], and chronic kidney disease [17]. A correlation has also been established between the risk of falls and the number of coexisting diseases [18,19].

Falls affect approximately 30% of the older population each year. They cause major injuries and reduce independence in daily functioning. The risk of falls increases in individuals who are less independent in basic and complex life activities [20]. Neelemaat et al. [21] conducted a randomized controlled trial that combined several components of nutritional care, including energy- and protein-enrichment of the diet, oral nutritional supplements, calcium and vitamin D supplementation, and telephone counseling. The intervention, which lasted from hospital admission to up to three months post-discharge, showed positive effects on energy and protein intake, vitamin D serum levels, and the incidence of falls.

According to the European Society of Clinical Nutrition and Metabolism (ESPEN), diagnosing malnutrition in older adults involves the use of the Full Mini Nutritional Assessment (F-MNA) questionnaire, which provides a comprehensive evaluation of the factors contributing to malnutrition [22]. In cases where the F-MNA is not feasible, simpler methods for assessing the risk of malnutrition or sarcopenia, such as measuring arm or calf circumference, may be used [23]. Regular monitoring of nutritional status in older adults allows early detection of appetite decline, malnutrition, and muscle mass reduction, enabling timely nutritional intervention. Preserving muscle mass and ensuring adequate nutrition are key to preventing falls.

This study aims to investigate the prevalence of malnutrition in different groups of older adults using the F-MNA, anthropometry, and s-albumin and the association between nutritional status and fall risk.

## 2. Materials and Methods

The study involved 228 participants (146 women and 82 men; median age, 72 years), aged from 60 to 93 years. The study protocol assumed one visit from a nurse, which took place at the hospital, a nursing care home, or participants’ homes. The nurse was professionally active, worked in the hospital (internal ward), and was trained for the nutritional status assessment.

The inclusion criteria for the study were:Age > 60 yearsSigned informed consentExpected survival time of more than 3 months or the absence of terminal illness

The project was approved by the Independent Ethics Committee of the Medical University of Gdańsk (NKEBN/376/2014), and all participants provided written informed consent.

### 2.1. Characteristics of the Study Population

Participants were recruited from three sources: hospitalized patients over the age of 60 in a university hospital, older individuals living at home, and residents of care homes. Most participants were married (95/228; 42%), 85 (37%) were widowed, and 32 (14%) were single.

Statistical analysis was performed in all study participants and additionally in three subgroups (separated based on place of stay), which initially did not differ significantly in terms of the number of participants, age, and BMI. Statistical calculations showed that the numbers of people in the subgroups were sufficient for further analysis.

Group 1: Patients hospitalized in the internal medicine ward (87 participants: 44 women, 43 men); the mean age was 71.60 ± 7.72, the median age was 70; and the mean BMI was 25.8 ± 4.6.

Group 2: People living at home (79 participants: 56 women, 23 men); the mean age was 70.05 ± 9.20, the median age was 68, and the mean BMI was 27.0 ± 4.2.

Group 3: Care home residents (62 participants: 46 women, 16 men); the mean age was 79.08 ± 7.33, the median age was 79.5, and the mean BMI was 26.9 ± 5.5.

Information on comorbidities and the number of medications taken was collected. The diseases diagnosed in the participants included coronary heart disease, persistent atrial fibrillation, hypertension, atherosclerosis, hypercholesterolemia, type 2 diabetes, obesity, rheumatoid arthritis, and depression.

### 2.2. Assessment of Nutritional Status and Appetite

#### 2.2.1. Anthropometric Measurements

Anthropometric measurements were collected by the same nurse during hospitalization, home visits, or care home visits. Body weight was measured using an electronic scale (Tanita BC420, Tanita Polska & Medkonsulting, Poznań, Poland), while height was measured to the nearest 5 mm using a stadiometer (SECA 213, Seca.gmbh & co., Hamburg, Germany). Body mass index (BMI) was calculated from weight and height (kg/m^2^) and classified according to World Health Organization criteria [22]. Arm (MAC), calf (CC), and waist circumferences (WC) were measured using a measuring tape.

Body composition was analyzed using bioelectrical impedance (Body Stat 1500, Body Stat LTD., London, UK). The parameters measured included fat mass (FM, kg), fat-free mass (FFM, kg), body weight, and total body water.

#### 2.2.2. F-MNA (Full Mini Nutritional Assessment)

The F-MNA was used to detect malnutrition or the risk of malnutrition in older adults. The screening assessment included questions on food intake, weight loss over the past 3 months, psychological stress, and BMI, followed by a comprehensive patient assessment. Scores on the F-MNA are categorized as follows: 24–30 points indicate normal nutritional status, 17–23.5 points indicate a risk of malnutrition, and scores below 17 indicate malnutrition.

#### 2.2.3. SNAQ (Simplified Nutritional Appetite Questionnaire)

The SNAQ was used to assess appetite and the risk of weight loss over the next 6 months. The questionnaire consists of four questions related to appetite loss, feelings of satiety, changes in taste, and the number of meals consumed per day. Responses are scored from A (1 point) to E (5 points). A total score of ≤14 indicates a significant risk of weight loss over the next 6 months.

#### 2.2.4. Fall Risk Assessment

The Tinetti scale, commonly used in comprehensive geriatric assessments, was employed to assess the risk of falls. It includes two sections: balance (maximum 16 points) and gait (maximum 12 points), for a total possible score of 28. A score below 26 suggests a risk of falling, while a score below 19 indicates a fivefold increased risk of falling compared to a score of 28 [24].

### 2.3. Biochemistry

Blood samples were collected after fasting and immediately analyzed. Serum albumin, triglycerides, total cholesterol, and HDL cholesterol were measured using standard laboratory methods.

### 2.4. Statistical Analysis

Statistical analysis was conducted using Statistica 13.0. Normality was tested using the Kolmogorov-Smirnov test. Differences between groups were analyzed using either Student’s *t*-test or the Mann-Whitney U test, as appropriate. The chi-squared test was used to examine the differences between the distributions of categorical variables. Data are presented as the mean ± standard deviation (SD) or as the medians and ranges. Correlations were analyzed using Spearman’s test. Stepwise multiple regression analysis was used to identify independent associations. Additionally, to determine the relationship between place of residence, nutritional status, and fall risk, an analysis was conducted in three subgroups that did not differ significantly in terms of age and BMI. A power analysis conducted in Statistica 13.0 indicated a required sample size of n = 63 (RMSSE = 0.25, power = 0.9, alpha = 0.05).

A *p*-value < 0.05 was considered statistically significant.

## 3. Results

### 3.1. Anthropometry, Nutritional Status, and Appetite

The average BMI across all study subjects was 26.53 ± 4.8; for women, it was 26.80 ± 5.17, and for men, it was 26.06 ± 4.05. The average waist circumference (WC) was elevated in all study groups. BMI did not differ significantly, and body composition analysis showed no significant differences in fat mass or fat-free mass between the three groups (see Table 1).

Calf circumference (CC) and mid-arm muscle circumference (MAMC) were significantly higher in individuals living at home than in hospitalized patients (33.07 ± 3.90 cm vs. 35.54 ± 4.23 cm, respectively; *p* < 0.05).

Statistically significant (*p* < 0.05) positive correlations were found between calf circumference and fat-free mass (FFM) (Spearman’s R = 0.43), mid-arm circumference (MAC) (Spearman’s R = 0.63), BMI (Spearman’s R = 0.62), and F-MNA score (Spearman’s R = 0.30).

#### 3.1.1. F-MNA

The results of the F-MNA are shown in Table 2. Across all participants, the average F-MNA score was 20.67 ± 5.02 points, indicating a risk of malnutrition. The highest percentage of malnourished individuals was in Group 1 (31 participants, 36%), while the lowest was in Group 2 (3 participants, 4%). In Group 3, 18 participants (29%) were malnourished (see Figure 1A).

Figure 1B,C show similar patterns for women (Figure 1B) and men (Figure 1C). The largest number of malnourished women was in Group 1 (16 women, 36%), whereas Group 2 had the highest proportion of women with a proper nutritional status (38 women, 68%). In Group 3, most women were at risk of malnutrition (27 women, 59%).

For men, the highest number of malnourished individuals was in Group 1 (15 men, 35%), and the lowest was in Group 2 (1 man, 4%). The highest risk of malnutrition for men was also observed in Group 1 (25 men, 58%). In Group 2, 15 men (65%) were properly nourished, while only 7 men (30%) were at risk of malnutrition.

The F-MNA score showed significant positive correlations with body weight (R = 0.35), BMI (r = 0.37), mid-arm circumference (MAC) (R = 0.36), calf circumference (CC) (R = 0.47), and serum albumin levels (R = 0.40).

#### 3.1.2. SNAQ

The SNAQ results (Table 2) indicate a significant risk of weight loss over the next 6 months across the study groups. Figure 2 shows the SNAQ scores by gender. A total of 146 participants (64%) were at risk of losing 5% of their body weight within the next 6 months. This risk was highest among individuals in the internal medicine ward (Group 1), with 75 participants (33%) at risk, particularly hospitalized men (39 men, 17%). In Group 3, women had a higher risk of weight loss (32 women, 14%). Among those living at home, the risk of losing 5% of body weight in the next 6 months was the lowest, affecting only 12% of the group (19 women, 8%, and 8 men, 3.5%).

### 3.2. The Tinnetti Scale

The Tinetti scale results indicated that the highest risk of falls was significantly more prevalent in Group 3, the care home residents, while the lowest risk was observed in those living at home (Table 2). Fall risk was a concern across all groups, as the overall scores were below 26 points. Additionally, in Group 3 (both men and women), the risk of falls was five times higher compared to individuals who scored 28 points on the Tinetti scale.

### 3.3. Comorbidity, Medication, and Biochemical Results

Group 2 had the lowest prevalence of diseases and medication use, with 49% of participants having more than 3 diseases and 71% taking more than 3 medications (Table 3). In Group 3, all participants were taking more than 3 prescription medications, and 84% had more than 3 diseases. Across Groups 1, 2, and 3, there were no statistically significant differences between genders regarding the number of diseases or medications taken.

Biochemical measurements showed that the mean serum albumin was 37.66 ± 7.56 g/L in Group 1, 41.37 ± 6.84 g/L in Group 2, and 31.05 ± 6.53 g/L in Group 3 (Table 4). There were statistically significant differences between these groups (*p* < 0.05). Group 3 had the lowest mean serum albumin level (31.05 ± 6.53 g/L), while in the other groups, albumin levels were within the normal range (35–55 g/L).

Regarding total cholesterol, none of the groups exceeded the normal limit (up to 200 mg/dL). Statistical differences were found between Group 2 (178 ± 40.91 mg/dL) and both Group 1 (172 ± 49.20 mg/dL) and Group 3 (157 ± 53.00 mg/dL), with Group 3 showing the lowest total cholesterol levels. In Group 1, the average HDL cholesterol concentration for both genders was 31.51 ± 12.74 mg/dL, with 31.62 ± 13.12 mg/dL in women and 31.20 ± 11.96 mg/dL in men. However, HDL cholesterol levels were too low across all groups.

As for triglycerides, normal levels were observed in all groups. Group 2 had the lowest average level (134 ± 58.26 mg/dL), while Group 3 had the highest (139 ± 48.55 mg/dL), though these differences were not statistically significant.

### 3.4. Regression Analysis and Correlations

The risk of falls, as assessed by the Tinetti scale, was significantly correlated with age (Spearman’s R = 0.5), calf circumference (Spearman’s R = 0.22), F-MNA score (Spearman’s R = −0.49), SNAQ score (Spearman’s R = −0.2), serum albumin (Spearman’s R = −0.31), the number of diseases (Spearman’s R = 0.4), and the number of medications taken daily (Spearman’s R = 0.41).

Regression analysis was performed in two models. Model I: dependent variable: fall risk, and independent variables: age, calf circumference, and residence in a nursing home. Model II: dependent variable: place of stay, and independent variables: fall risk, s-albumin, and CC. Regression analysis identified age, calf circumference, and residence in a nursing home as independent predictors of fall risk among older adults (Table 5).

## 4. Discussion

The results of our study indicate a high risk of malnutrition among different groups of older adults, especially during hospitalization and long-term stays in nursing care homes. Malnutrition was associated with comorbidities, the number of medications taken, and decreased appetite. Previous studies have established a relationship between immobility, such as during hospitalization, and muscle mass and function, which in turn increases the risk of falls. In our study, statistical analysis demonstrated associations between the risk of falls (as measured by the Tinetti scale) and factors including age, calf circumference, appetite, comorbidities, the number of medications, and nutritional status as assessed by the MNA, and serum albumin levels.

### 4.1. Nutritional Status

Despite the relatively high BMI and waist circumference in the study population, approximately 26% were malnourished or at risk of malnutrition. Furthermore, 64% of participants were at risk of losing 5% of their body weight in the next 6 months. While BMI is widely used to assess the risk of overweight and obesity, it has limitations, particularly in older populations. Body composition changes with age; aging is associated with a decrease in muscle mass and strength and an increase in fat mass, especially in the abdominal area, which may contribute to an increased risk of sarcopenia [25]. Therefore, assessing nutritional status should not rely solely on BMI, as this measure does not account for age-related differences in lean and fat mass [26,27]. Although BMI can be a strong predictor of obesity-related health issues, it may under-detect the risk of malnutrition in overweight and obese patients [28]. Recent studies have emphasized that muscle mass and its quality are crucial for overall health in older adults. Assessing body composition using bioelectrical impedance analysis (BIA) or dual-energy X-ray absorptiometry (DXA) is recommended for measuring fat-free mass. However, in situations where these diagnostic tests are not feasible (e.g., in patients with pacemakers or no access to DXA), reference anthropometric data, such as calf circumference, may be essential for assessing muscle mass in older adults [22].

It is worth noting that discharge from the hospital and subsequent long-term care play a significant role in the recovery of older individuals. Hospitalization significantly affects body composition; even in healthy adults, immobilization can lead to a loss of approximately 1.4 kg of muscle tissue per week, and illness and stress can exacerbate this process. Inadequate nutrition also negatively impacts the body; studies have shown that during hospitalization, 60% of patients consume less than half of their meals, and hospital food is not always tailored to the needs of older patients [29].

In the study by Pavlovic et al. [30], similar to our findings, community-dwelling older adults exhibited higher BMI (27.94 ± 4.73), weight (78.40 ± 13.99), waist circumference (97.77 ± 12.88), hip circumference (104.61 ± 11.47), upper arm circumference (27.54 ± 3.58), and calf circumference (34.02 ± 5.16) values than those living in nursing homes. The results of this study and others suggest that staff involved in the care of older adults in nursing facilities should implement routine malnutrition screening as part of comprehensive geriatric assessments.

In this context, assessing patients’ nutritional status (using tools like the F-MNA and SNAQ) and conducting biochemical tests (such as serum albumin levels) may be helpful. Hypoalbuminemia is common in hospitalized patients and is associated with adverse clinical outcomes. Increased mortality rates have been documented in patients with low serum albumin levels in both hospital and community settings. While serum albumin is widely used as a diagnostic marker of malnutrition in clinical practice, it should not be used alone, as both inflammation and nutritional risk contribute to low serum albumin levels in acutely ill patients [31].

### 4.2. Appetite

Decreased appetite is one symptom of deteriorating health. Therefore, it is crucial to implement appropriate tools in everyday clinical practice that allow quick and easy identification of this symptom. Recognizing loss of appetite is an early indicator of malnutrition, making it essential to monitor this factor during a patient’s admission, such as during hospitalization and after discharge [32,33].

The SNAQ results indicate decreased appetite and a significant risk of weight loss in the next six months among the study population. Most individuals at risk of weight loss were hospitalized in the internal medicine ward and lived alone. Researchers Krzymińska-Siemaszko et al. [1] also reached similar conclusions, finding that the strongest socio-economic predictor of poor nutritional status among older Poles living in the community was subjective loneliness. Eskelinen et al. [34] identified various mechanisms by which loneliness may lead to decreased appetite, including low mood, deterioration of physical fitness, loss of social relationships, and subsequent isolation at home. In the study by Ramic et al. [35], differences in nutritional status were observed between older adults living alone (study group) and those living in family environments (control group). Individuals in the study group had lower BMI values compared to the control group and were at an increased risk of malnutrition due to a reduction in daily meal frequency, which in turn lowered their consumption of protein, vegetables, and fruit. The study group also reported loss of appetite and rated their health as worse compared to others.

### 4.3. Risk of Falls

The Tinetti scale showed that the highest risk of falls was characteristic of residents in care homes, while the lowest risk was observed among those living in family homes. A risk of falls was noted in all study groups, as the overall scores were below 26 points. Malnutrition significantly influences the risk of falls, contributing to decreased muscle strength [36]. Similar findings were reported in hospitalized patients in the study by Adly et al. [37]. Also, Eckert et al. suggest that falls are associated with malnutrition risk in community-dwelling older adults [38].

### 4.4. Comorbidity and Polypharmacy

Another issue related to nutritional status in the older population is comorbidity and polypharmacy. Gazzotti et al. [39] recorded the number of medications taken by each patient and explored its relationship with nutritional status. A significant correlation (*p* = 0.014) was found between the number of medications prescribed and the number of diseases. In that study, low F-MNA scores were frequently associated with the consumption of laxatives (*p* = 0.001), sedatives (*p* < 0.001), and steroids (*p* = 0.017). No significant association was found between F-MNA scores and other medications, such as antibiotics or thyroid hormones. Our study also revealed a significant correlation (−0.36) between the number of medications and the F-MNA score, as well as between the number of medications and the SNAQ score (−0.32). Furthermore, malnutrition may potentiate the effects of certain medications, leading to adverse effects by altering their pharmacokinetic and pharmacodynamic properties, which can increase the patient’s sensitivity to the drug [40]. Consequently, the drug may exert a stronger effect at the usual dose, heightening the occurrence of side effects [41]. Another aspect of long-term medication use is the potential for anorexia and gastrointestinal disorders. Taking multiple medications can alter sensory perception of taste and affect nutrient absorption and metabolism [42,43].

The majority of study individuals suffered from hypertension (58.11%), which was strongly related to BMI (R = 0.62, *p* < 0.05). In our study, ischemic heart disease and atrial fibrillation were most frequently observed in Group 3 (20.97% for both). Type 2 diabetes was also common, affecting 38.16% of all surveyed individuals. Significant differences were noted between Groups 1 and 3 and Group 2, with type 2 diabetes being most prevalent in Groups 1 (41.38%) and 3 (40.32%).Numerous studies indicate a relationship between malnutrition in older people and cardiovascular diseases [9,44,45,46,47]. In the study by Junaida et al. [48], according to the MNA-SF, malnutrition and the risk of malnutrition were significantly more common in older adults with type 2 diabetes. While our study did not find a strong correlation between F-MNA and SNAQ results and type 2 diabetes, a significant correlation was observed between depression and F-MNA scores (−0.46), as well as between depression and SNAQ scores (−0.30). Similar conclusions were reached by Simões et al. [49]. A strong relationship between the severity of malnutrition and depressive symptoms was also found in the study by Cansel and Yakaryilmaz et al. [50], where the prevalence of depressive symptoms was 62.7% in malnourished patients compared to 42.3% in those at risk of malnutrition.

Our study presented some limitations. A relatively small group of people was studied, but the advantage was that it included three uniform groups, which allowed their comparison in terms of nutritional status, in turn emphasizing the goals of care for the elderly in these three situations—in a family home, a nursing home, and a hospital. The study may indicate the need for the organization of care for elderly people in these three situations. In addition, it was an advantage to examine the body compositions of all the people studied. Another limitation is the lack of long-term observation and performing the tests at one time point. However, our study indicates the need for a comprehensive assessment of older people, which is not always carried out in practice.

Overall, our findings suggest that malnutrition is closely associated with multi-morbidity and an increased risk of falls, both of which are major causes of disability in older adults. Importantly, the risk of both malnutrition and falls was linked to low muscle mass, as indicated by low calf circumference, independent of BMI.

## 5. Conclusions

Our research demonstrates that older adults are at risk of malnutrition, particularly during hospitalization and in long-term stays in nursing care homes. Hospitalized individuals exhibited the poorest nutritional status and were at significant risk of further weight loss, underscoring the importance of post-discharge care and rehabilitation. Attention should be focused on the post-hospitalization recovery period to prevent exacerbation of malnutrition and falls. Appropriate rehabilitation and tailored dietary interventions during this time can reduce the risk of adverse outcomes and improve the overall well-being of older adults. Consequently, it is necessary to implement appropriate diagnostic methods for malnutrition, including anthropometric measurements, F-MNA, SNAQ, and laboratory tests.

Clinical implications. Our study indicates the need to assess the nutritional status of older people, shows the relationship between malnutrition and falls, and indicates the need for comprehensive care for older people. Further research is indicated, in particular, to assess the health status of older people after discharge from the hospital and during longer periods of care in nursing homes and to assess risk factors for malnutrition in people staying at home by assessing their diet and the need for assistance from other people.

## Figures and Tables

**Figure 1 nutrients-16-03694-f001:**
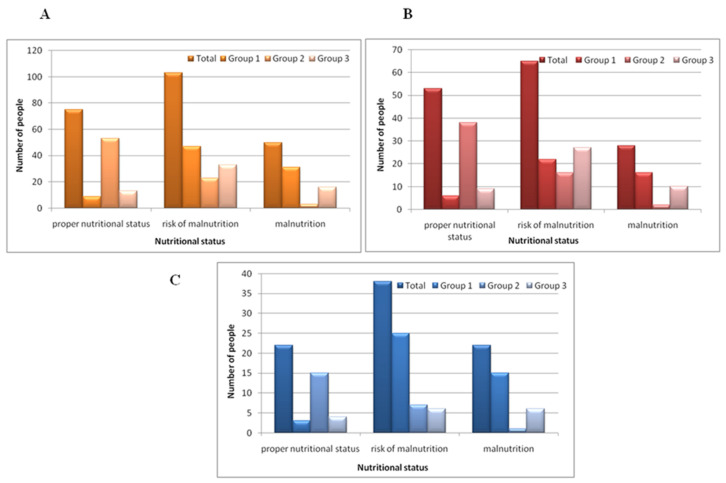
F-MNA results in all groups (n = 228) (**A**), in all women (n = 146) (**B**), and in all men (n = 82) (**C**). Group 1—internal medicine ward patients, Group 2—family home residents, Group 3—care home residents.

**Figure 2 nutrients-16-03694-f002:**
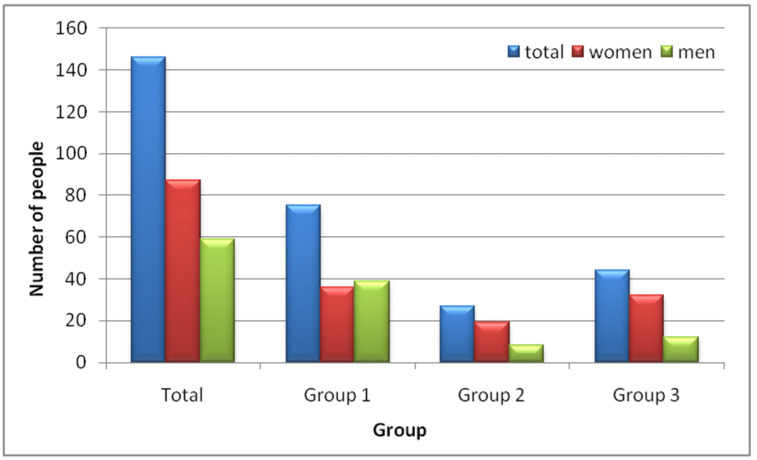
SNAQ results. Group 1—internal medicine ward patients, Group 2—family home residents, Group 3—care home residents.

**Table 1 nutrients-16-03694-t001:** Anthropometric measurements and body composition in the study groups.

Group	Sex	Group Size	FM [kg]	FFM [kg]	TBW [%]	BMI [kg/m^2^]	Arm Circumference [cm]	Calf Circumference [cm]	Waist Circumference [cm]
Total	total	228	25.17 ± 9.36 ^b^	46.41 ± 11.21 ^b^	53.74 ± 8.36 ^b^	26.53 ± 4.80 ^a^	24.50 ± 4.54 ^a^	34.08 ± 4.29 ^ab^	96.71 ± 14.34 ^b^
	women	146	27.28 ± 9.60 ^c^	40.86 ± 7.05 ^a^	51.29 ± 7.89 ^a^	26.80 ± 5.17 ^a^	28.45 ± 4.67 ^ab^	33.90 ± 4.38 ^ab^	95.76 ± 14.89 ^b^
	men	82	21.42 ± 7.63 ^ab^	56.85 ± 9.86 ^c^	58.09 ± 7.39 ^bc^	26.06 ± 4.05 ^a^	29.38 ± 4.26 ^ab^	34.40 ± 4.12 ^ab^	98.42 ± 13.21 ^b^
Group 1	total	87	23.96 ± 8.83 ^b^	48.01 ± 11.37 ^b^	54.63 ± 8.40 ^b^	25.81 ± 4.66 ^a^	27.89 ± 4.81 ^a^	33.07 ± 3.90 ^a^	94.73 ± 13.77 ^b^
	women	44	26.65 ± 10.00 ^bc^	40.19 ± 6.65 ^a^	51.97 ± 8.31 ^a^	26.12 ± 5.63 ^a^	27.13 ± 5.43 ^a^	32.67 ± 4.20 ^a^	92.19 ± 15.16 ^ab^
	men	43	21.19 ± 6.46 ^ab^	56.02 ± 9.46 ^c^	57.36 ± 7.66 ^bc^	25.49 ± 3.46 ^a^	28.67 ± 3.99 ^ab^	33.56 ± 3.54 ^ab^	97.44 ± 11.75 ^b^
Group 2	total	79	25.81 ± 8.40 ^bc^	46.94 ± 11.27 ^b^	52.66 ± 6.97 ^ab^	27.03 ± 4.21 ^a^	29.47 ± 4.33 ^ab^	35.54 ± 4.23 ^b^	97.20 ± 12.82 ^b^
	women	56	27.82 ± 8.30 ^c^	41.87 ± 8.07 ^a^	50.42 ± 6.34 ^a^	27.25 ± 4,43 ^a^	29.09 ± 3.84 ^ab^	35.54 ± 4.17 ^b^	96.36 ± 13.32 ^b^
	men	23	20.92 ± 6.77 ^ab^	59.29 ± 7.90 ^cd^	58.10 ± 5.32 ^bc^	26.52 ± 3.67 ^a^	30.41 ± 5.32 ^ab^	35.57 ± 4.46 ^b^	99.24 ± 11.53 ^b^
Group 3	total	62	26.06 ± 11.10 ^bc^	44.21 ± 10.76 ^ab^	53.85 ± 9.80 ^b^	26.90 ± 5.59 ^a^	29.16 ± 4.27 ^ab^	33.63 ± 4.45 ^ab^	98.89 ± 16.64 ^b^
	women	46	27.22 ± 10.79 ^c^	40.26 ± 6.02 ^a^	51.70 ± 9.16 ^a^	26.90 ± 5.59 ^a^	28.93 ± 4.65 ^ab^	33.15 ± 4.27 ^ab^	98.54 ± 16.01 ^b^
	men	16	22.73 ± 11.32 ^ab^	55.58 ± 13.02 ^c^	60.04 ± 9.15 ^c^	26.90 ± 5.77 ^a^	29.81 ± 2.90 ^ab^	35.00 ± 4.80 ^ab^	99.88 ± 18.86 ^b^

Group 1—internal medicine ward patients, Group 2—family home residents, Group 3—care home residents; parameters are presented as the mean ± standard deviation. Values with the same letter (^a,b,c,d^) are not significantly different, while values with different letters are statistically different (*p* < 0.05).

**Table 2 nutrients-16-03694-t002:** Results of the nutritional status assessment (F-MNA) and the risk of falls.

Group	Sex	Group Size	F-MNA			SNAQ	Tinneti
			Screening Score (Points)	Assessment (Points)	Final Score (Points)	Score (Points)	Score (Points)
Total	total	228	9.90 ± 3.10 ^b^	10.82 ± 2.49 ^b^	20.67 ± 5.02 ^bc^	13.25 ± 3.09 ^ab^	20.06 ± 6.17 ^b^
	women	146	10.12 ± 3.13 ^b^	10.95 ± 2.34 ^b^	21.09 ± 4.93 ^bc^	13.42 ± 3.04 ^ab^	19.64 ± 6.35 ^b^
	men	82	9.50 ± 3.01 ^b^	10.57 ± 2.74 ^ab^	19.94 ± 5.12 ^ab^	12.94 ± 3.16 ^ab^	20.79 ± 5.81 ^b^
Group 1	total	87	8.72 ± 2.74 ^a^	9.63 ± 2.17 ^a^	18.35 ± 4.33 ^a^	11.72 ± 3.14 ^a^	20.54 ± 5.43 ^b^
	women	44	8.93 ± 2.97 ^a^	9.86 ± 2.29 ^a^	18.80 ± 4.64 ^a^	11.77 ± 3.28 ^a^	20.07 ± 5.58 ^b^
	men	43	8.51 ± 2.82 ^a^	9.39 ± 2.83 ^a^	17.90 ± 5.66 ^a^	11.67 ± 3.07 ^a^	21.02 ± 5.28 ^b^
Group 2	total	79	11.72 ± 2.49 ^c^	12.72 ± 1.81 ^c^	24.49 ± 3.55 ^c^	15.05 ± 2.42 ^c^	23.19 ± 5.48 ^c^
	women	56	11.59 ± 2.63 ^c^	12.91 ± 1.53 ^c^	24.31 ± 3.56 ^c^	14.91 ± 2.60 ^c^	22.75 ± 5.98 ^bc^
	men	23	8.79 ± 2.10 ^a^	9.64 ± 2.34 ^a^	18.43 ± 3.53 ^a^	11.79 ± 1.92 ^a^	24.26 ± 3.92 ^c^
Group 3	total	62	9.23 ± 3.25 ^b^	9.82 ± 2.39 ^a^	19.07 ± 4.72 ^ab^	13.08 ± 2.56 ^ab^	15.39 ± 5.16 ^a^
	women	46	9.48 ± 3.20 ^b^	9.84 ± 1.88 ^a^	19.35 ± 4.61 ^ab^	13.17 ± 2.42 ^ab^	15.46 ± 5.13 ^a^
	men	16	8.5 ± 3.37 ^a^	9.78 ± 2.04 ^a^	18.28 ± 5.09 ^a^	12.81 ± 3.02 ^ab^	15.19 ± 5.42 ^a^

Group 1—internal medicine ward patients, Group 2—family home residents, Group 3—care home residents; parameters are presented as the mean ± standard deviation. Values with the same letter (^a,b,c^) are not significantly different, while values with different letters are statistically different (*p* < 0.05).

**Table 3 nutrients-16-03694-t003:** Numbers of people suffering from selected diseases.

Group	Sex	Group Size	Coronary Heart Disease	Persistent Atrial Fibrillation	Hypertension	Atherosclerosis	Hypercholesterolemia	Type 2 Diabetes	Obesity	Rheumatoid Arthritis (RA)	Depression
Total	total	228	32 (14.04) ^bc^	29 (12.72) ^b^	134 (58.77) ^b^	17 (7.46) ^a^	44 (19.30) ^bc^	87 (38.16) ^b^	45 (19.74) ^ab^	45 (19.74) ^bc^	32 (14.04) ^b^
	women	146	18 (12.33) ^b^	21 (14.38) ^b^	89 (60.96) ^b^	10 (6.85) ^a^	32 (21.92) ^c^	56 (38.36) ^b^	30 (20.55) ^b^	39 (26.71) ^c^	20 (13.70) ^b^
	men	82	14 (17.07) ^c^	8 (9.76) ^a^	44 (53.66) ^a^	7 (8.54) ^a^	12 (14.63) ^b^	31 (37.80) ^b^	15 (18.29) ^ab^	6 (7.32) ^a^	12 (14.63) ^b^
Group 1	total	87	11 (12.64) ^b^	7 (8.05) ^a^	48 (55.17) ^ab^	4 (4.60) ^a^	18 (20.69) ^c^	36 (41.38) ^c^	16 (18.39) ^ab^	13 (14.94) ^b^	13 (14.94) ^b^
	women	44	3 (6.82) ^a^	4 (9.09) ^a^	26 (59.10) ^b^	2 (4.55) ^a^	11 (25.00) ^c^	18 (40.91) ^c^	9 (20.45) ^b^	11 (25.00) ^c^	7 (15.91) ^b^
	men	43	8 (18.60) ^c^	3 (6.98) ^a^	22 (51.16) ^a^	2 (4.65) ^a^	7 (16.28) ^b^	18 (41.86) ^c^	7 (16.28) ^ab^	2 (4.65) ^a^	6 (13.95) ^b^
Group 2	total	79	8 (10.13) ^ab^	9 (11.39) ^b^	44 (55.70) ^ab^	3 (3.80) ^a^	15 (18.99) ^b^	26 (32.91) ^a^	14 (17.72) ^ab^	14 (17.72) ^b^	5 (6.33) ^a^
	women	56	7 (12.50) ^b^	5 (8.93) ^a^	32 (57.14) ^b^	2 (3.57) ^a^	13 (23.21) ^c^	18 (32.14) ^a^	11 (19.64) ^ab^	12 (21.43) ^bc^	4 (7.14) ^a^
	men	23	1 (4.35) ^a^	4 (17.39) ^bc^	11 (47.83) ^a^	1 (4.35) ^a^	2 (8.70) ^a^	8 (34.78) ^a^	3 (13.04) ^a^	2 (8.70) ^a^	1 (4.35) ^a^
Group 3	total	62	13 (20.97) ^d^	13 (20.97) ^bc^	42 (67.74) ^c^	10 (16.13) ^b^	11 (17.74) ^b^	25 (40.32) ^c^	15 (24.19) ^c^	18 (29.03) ^c^	14 (22.58) ^b^
	women	46	8 (17.39) ^c^	12 (26.09) ^c^	31 (67.39) ^c^	6 (13.04) ^b^	8 (17.39) ^b^	20 (43.48) ^c^	10 (21.74) ^b^	16 (34.78) ^d^	9 (19.57) ^b^
	men	16	5 (31.25) ^e^	1 (6.25) ^a^	11 (68.75) ^c^	4 (25.00) ^c^	3 (18.75) ^b^	5 (31.25) ^a^	5 (31.25) ^d^	2 (12.50) ^b^	5 (31.25) ^c^

Group 1—internal medicine ward patients, Group 2—family home residents, Group 3—care home residents; parameters are presented as the mean ± standard deviation. Values with the same letter (^a,b,c,d,e^) are not significantly different, while values with different letters are statistically different (*p* < 0.05).

**Table 4 nutrients-16-03694-t004:** Health conditions of the study participants.

Group	Sex	Group Size	Number of Diseases [n (%)]		Number of Medications [n (%)]		Biochemical Results			
			≤3	4 and More	≤3	4 and More	s-Albumin [g/L]	Total Cholesterol [mg/dL]	Triglycerides [mg/dL]	HDL Cholesterol [mg/dL]
Total	total	228	77 (33.77) ^b^	151 (66.23) ^b^	34 (14.91) ^b^	194 (85.09) ^b^	37.14 ± 8.24 ^bc^	170 ± 48.12 ^b^	135 ± 54.53 ^a^	37.13 ± 20.38 ^ab^
	women	146	46 (31.51) ^b^	100 (68.49) ^b^	23 (15.75) ^b^	123 (84.25) ^b^	37.24 ± 8.17 ^bc^	177 ± 44.88 ^b^	139 ± 55.43 ^a^	47.65 ± 15.27 ^ab^
	men	82	30 (36.59) ^b^	52 (63.41) ^b^	11 (13.41) ^b^	71 (86.59) ^b^	36.98 ± 8.41 ^bc^	158 ± 51.54 ^ab^	129 ± 52.52 ^a^	36.21 ± 27.31 ^ab^
Group 1	total	87	27 (31.03) ^b^	60 (68.97) ^b^	11 (12.64) ^b^	76 (87.36) ^b^	37.65 ± 7.95 ^b^	172 ± 49.20 ^b^	135 ± 55.55 ^a^	38.18 ± 28.05 ^ab^
	women	44	12 (27.27) ^b^	32 (72.73) ^b^	7 (15.91) ^b^	37 (84.09) ^b^	37.66 ± 7.56 ^b^	183 ± 39.77 ^b^	135 ± 51.00 ^a^	37.79 ± 16.81 ^ab^
	men	43	14 (32.56) ^b^	29 (67.44) ^b^	4 (9.30) ^b^	39 (90.70) ^b^	37.64 ± 8.42 ^b^	161 ± 55.43 ^ab^	134 ± 60.47 ^a^	38.58 ± 36.36 ^ab^
Group 2	total	79	40 (50.63) ^c^	39 (49.37) ^a^	23 (29.11) ^c^	56 (70.89) ^a^	41.37 ± 6.84 ^c^	178 ± 40.91 ^b^	134 ± 58.26 ^a^	40.40 ± 13.32 ^ab^
	women	56	26 (46.43) ^c^	30 (53.57) ^a^	16 (28.57) ^c^	40 (71.43) ^a^	41.86 ± 6.40 ^c^	186 ± 40.47 ^b^	140 ± 63.21 ^a^	42.01 ± 14.11 ^ab^
	men	23	14 (60.87) ^d^	9 (39.13) ^a^	7 (30.43) ^c^	16 (69.57) ^a^	39.57 ± 7.62 ^bc^	162 ± 38.96 ^ab^	122 ± 45.31 ^a^	35.29 ± 9.71 ^ab^
Group 3	total	62	10 (16.13) ^a^	52 (83.87) ^c^	0 (0.00) ^a^	62 (100) ^c^	31.05 ± 6.53 ^a^	157 ± 53.00 ^a^	139 ± 48.55 ^a^	31.51 ± 12.74 ^a^
	women	46	8 (17.39) ^a^	38 (82.61) ^c^	0 (0.00) ^a^	46 (100) ^c^	30.91 ± 6.30 ^a^	161 ± 51.38 ^ab^	145 ± 50.86 ^a^	31.62 ± 13.12 ^a^
	men	16	2 (12.50) ^a^	14 (87.50) ^c^	0 (0.00) ^a^	16 (100) ^c^	31.47 ± 7.35 ^a^	147 ± 43.13 ^a^	122 ± 37.91 ^a^	31.20 ± 11.96 ^a^

Group 1—internal medicine ward patients, Group 2—family home residents, Group 3—care home residents. The age value is the mean ± standard deviation. Values with the same letter (^a,b,c,d^) are not significantly different, while values with different letters are statistically different (*p* < 0.05).

**Table 5 nutrients-16-03694-t005:** Multivariate regression model predicting the fall risk (the adjusted R^2^ of the model was 0.27; *p* < 0.000).

Regression Model	Beta	B	Standard Error	*p*-Value
Constant		−0.93	0.39	0.01
Age	0.56	0.02	0	0
Residence (care home)	0.36	0.24	0.04	0
F-MNA	0.09	0	0	0.05
Calf circumference	−0.17	−0.02	0.01	0
Gender	0.01	0.02	0.08	0.75
BMI	−0.02	−0.00	0.01	0.74

## Data Availability

Data is contained within the article.

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
