# Peer review of "Malnutrition and Fall Risk in Older Adults: A Comprehensive Assessment Across Different Living Situations"

_nutrients, 2024, doi:10.3390/nu16213694_

Round 1
Reviewer 1 Report
Comments and Suggestions for Authors
The work carried out by Mziray et al. needs some revisions.
The abstract should not be structured.
Line 16: Which are these diagnostic methods?
You should mention directions for further investigations and what the readers can learn from your study.
The sample size seems not representative of the study population. Can you please justify this in your manuscript and explain how you consider adequate the number of 228 participants?
Line 103: Which electronic scale? Provide details of it and of all the used equipment in your research.
Why do you have only 3 lines on page 5? Page 6 is not numbered and the numbering of the following pages is not correct.
In the Results section, tables need a better explanation. The format of the pages needs to be adjusted.
Lines 284-297: References are missing. Compare the obtained results you mentioned with other studies.
I suggest you elaborate on your study’s limitations and strengths.
The Conclusions should be aligned with the revisions you will make in the abstract.
Author Response
Thank you very much for your response. We feel that the Reviewer’s comments are valuable and contribute to the improvement of our article. We have reviewed the manuscript and performed the suggested improvements. Our answers to the Reviewer’s points and the consequent changes are described in detail below.
Answers for the Reviewer’s comments:
Comment . The abstract should not be structured.
Answer: We have corrected the abstract according to the recommendation
Malnutrition among older adults is associated with numerous adverse effects, including increased morbidity, mortality, prolonged hospital stays, and a heightened risk of falls. This study aims to investigate the prevalence of malnutrition in different groups of older adults using F-MNA, anthropometry and s-albumin and the association between nutritional status and fall risk. A total of 228 participants aged 60 years and older were divided into three groups: (1) patients in an internal medicine ward, (2) individuals living in family homes, and (3) residents of care homes. Disease profiles, nutritional status (assessed using F-MNA and SNAQ), body composition, fall risk, and biochemical markers were evaluated. The results indicated the highest prevalence of malnutrition among hospitalized individuals. Fall risk was associated with age, calf circumference, F-MNA, SNAQ, serum albumin levels, residence in a care home, comorbidities, and the number of medications taken daily. Regression analysis revealed that age, calf circumference, and residence in a care home were independent predictors of fall risk in older adults. Older adults are at significant risk of malnutrition, with the risk notably increasing during hospitalization and long-term stays in care homes. Hospitalized individuals had the poorest nutritional status and were at significant risk of further weight loss, underscoring the importance of post-discharge care and rehabilitation.
Comment: Line 16: Which are these diagnostic methods?
Answer: We have provided more details: This study aims to investigate the prevalence of malnutrition in different groups of older adults using F-MNA, anthropometry and s-albumin.
Comment : You should mention directions for further investigations and what the readers can learn from your study.
Answer: Our study indicates the need to assess the nutritional status of older people, shows the relationship with malnutrition and falls, and indicates the need for comprehensive care for older people. Further research is indicated, in particular, to assess the health status of older people after discharge from hospital, during longer periods of care in nursing homes, and to assess risk factors for malnutrition in people staying at home by assessing their diet and the need for assistance from other people.
Comment: The sample size seems not representative of the study population. Can you please justify this in your manuscript and explain how you consider adequate the number of 228 participants?
Answer: In fact, it would be better if the study group was larger, but it included all elderly people who agreed and had no contraindications to undergo electrical bioimpedance testing (e.g. a cardiac pacemaker) and demonstrated the ability to perform the F-MNA and Tinnetti scale. The study design assumed a minimum of 100 people in each group.
Comment Line 103: Which electronic scale? Provide details of it and of all the used equipment in your research.
Answer: We have completed the data in the methods section. Body weight was measured using an electronic scale (Tanita BC420 ), while height was measured to the nearest 5 mm using a stadiometer (SECA 213, Hamburg, Germany). Body composition was analyzed using bioelectrical impedance (Body Stat 1500, Limited, UK).
Comment: Why do you have only 3 lines on page 5? Page 6 is not numbered and the numbering of the following pages is not correct.
In the Results section, tables need a better explanation. The format of the pages needs to be adjusted.
Answer: We have improved text formatting, we added additional information to description of the Table.
Comment Lines 284-297: References are missing. Compare the obtained results you mentioned with other studies.
Answer: We added missing reference
Comment I suggest you elaborate on your study’s limitations and strengths.
Answer: Acording Reviewer’s suggastion we added: limitation and strengths of the study
Our study presented some limitations it was the relatively small group of people studied, but the advantage was that it included 3 uniform groups, which allowed for their comparison in terms of nutritional status and emphasis on the goals of care for the elderly in these three situations - in a family home, a nursing home and in a hospital. The study may indicate the needs of the organization of care for the elderly in these three situations. In addition, it was an advantage to examine the body composition of all the people studied. Another limitation is the lack of long-term observation and performing the tests at one time point. However, our study indicates the need for a comprehensive assessment of older people, which is not always done in practice.
Comment The Conclusions should be aligned with the revisions you will make in the abstract.
Answer: We corrected Abstract and Conclusion

Reviewer 2 Report
Comments and Suggestions for Authors
this is a very interesting research, but it has gaps in methodology, you perform and observational cross over study, but you divide your patients into 3 groups according to recrutiment origin, to test differecnes you have to ensure there are no baseline differecnes between your patients, so you hass follow a deisgn type case-control or perform a propensity score matching to extract a comparable subsample. On the ohter hand, in this type of study is better to perform regression analysis and, finally, you have to justify your sample size calculation
Author Response
Answer to Reviewer.
Thank you very much for Reviewer’s comments. We agree with the Reviewer, that an observational (nonrandomized) study does not use random allocation to assign participants to comparison groups. We are aware, it is difficult to ensure that the comparison groups in nonrandomized studies are sufficiently similar especially with respect to prognostic variables. However, we undertook a comparison of the three separate groups because they were initially similar in terms of number of people, age and BMI (we added information in section Characteristics of the Study Population).
We additionally performed regression analysis, dependent variable: place of residence, independent variables: risk of falls, s-albumin, calf circumference, which showed a statistically significant relationship between the dependent variable and the set of independent variables.
Power analysis of the test performed in the Statistica 13.0 indicated the group size n= 63 (RMSSE 0.25, power 0.9, alpha = 0.05).

Round 2
Reviewer 1 Report
Comments and Suggestions for Authors
Dear authors, I’m not satisfied with this answer:
Comment: The sample size seems not representative of the study population. Can you please justify this in your manuscript and explain how you consider adequate the number of 228 participants?
Answer: In fact, it would be better if the study group was larger, but it included all elderly people who agreed and had no contraindications to undergo electrical bioimpedance testing (e.g. a cardiac pacemaker) and demonstrated the ability to perform the F-MNA and Tinnetti scale. The study design assumed a minimum of 100 people in each group.
Why did you assume a minimum of 100 people in each group? How do you justify this? There are statistical analyses to verify the representativeness of your sample. I believe this is crucial to give to your study the important impact that deserves a paper published in a Q1 international journal.
Author Response
Answer to Reviewer.
Thank you very much for Reviewer’s comments. Just In response to the second Reviewer, we presented the results of the power analysis. Power analysis of the test performed in the Statistica 13.0 indicated the group size n= 63 (RMSSE 0.25, power 0.9, alpha = 0.05).
We planned to examine more than calculated number of people (100 people from three places of stay), knowing the problems with drop-out (contraindications to bioimpedance, possibility of performing tests, mental state of the subjects, lack of material for laboratory analysis) based on our previous studies, e.g. POLSENIOR (Socioeconomic Risk Factors of Poor Nutritional Status in Polish Elderly Population: The Results of PolSenior2 Study. Krzymińska-Siemaszko R, Deskur-Śmielecka E, Kaluźniak-Szymanowska A, Kaczmarek B, Kujawska-Danecka H, Klich-Rączka A, Mossakowska M, Małgorzewicz S, Dworak LB, Kostka T, Chudek J, Wieczorowska-Tobis K. Nutrients. 2021 Dec 8;13(12):4388. doi: 10.3390/nu13124388.)
We have also improved the Material and Methods sections.
Reviewer 2 Report
Comments and Suggestions for Authors
paper is now right for pubblication
Author Response
Thank you for accepting our replies and correctionsRound 3
Reviewer 1 Report
Comments and Suggestions for Authors
The power analysis has to be reflected in the manuscript.
Author Response
Thank you for the Reviewer’s comments. We have decided to add the following information to the Statistical Analysis section:
"Additionally, to determine the relationship between place of residence, nutritional status, and fall risk, an analysis was conducted in three subgroups that did not differ significantly in terms of age and BMI. A power analysis conducted in Statistica 13.0 indicated a required sample size of n = 63 (RMSSE = 0.25, power = 0.9, alpha = 0.05)."